



# Analyze and improve the influence of geomagnetic gradient on aeromagnetic compensation in a towed bird

Zhijian Zhou[1,2], Zhilong Liu[1,2], Wenduo Li[1,2], Yihang Wang[1,2], and Chao Wang[1,2]

[1]Key Laboratory of Geo-exploration Instruments, Ministry of Education of China, Jilin University, Changchun 130026, China
[2]College of Instrumentation and Electrical Engineering, Jilin University, Changchun 130026, China
*Correspondence to*: Chao Wang (chaow@jlu.edu.cn)

**Abstract.** Aeromagnetic exploration is an important method of geophysical exploration. We study the compensation method of towed bird system and establish the towed bird interference model. Due to the low altitude of the helicopter, the geomagnetic gradient changes greatly, so the geomagnetic gradient is considered in the towed bird interference model. In this paper, we model the gradient of the geomagnetic field as vertical gradient and horizontal gradient components, analyze the influence of the towed bird system on the compensation results under different motion modes, and apply the ridge estimation method to solve the problem. We verify the feasibility of this compensation method through actual flight tests, and further improve the data quality of the towed bird interference.

## 1 Introduction

Aeromagnetic exploration is an important means of geological research and material exploration(Nabighian et al., 2005). Since the magnetic field generated by the ferromagnetic material and metal cutting geomagnetic field lines on the aircraft platform will interfere with the magnetic detector, and then affect the quality of the aeromagnetic detection data, it is necessary to carry out aeromagnetic compensation.

In 1950, Tolles and Lawson summarized three sources related to aircraft maneuvers: the permanent field, the induced field, and the eddy current field (Tolles and Lawson, 1950). In 1961, Leliak summarized the work of Tolles and Lawson and proposed a model of aeromagnetic compensation called the Tolles Lawson (T-L) model (Leliak, 1961). As a linear solution, the T-L model faces the problem of multicollinearity (Leach, 1979; Bickel, 1979). In 1979, Bickle analyzed the multicollinearity of the T-L model and proposed a small signal solving method to reduce the linear relationship between features (Bickel, 1979). In 1980, Leach used linear regression theory to study the T-L model and proposed a ridge regression algorithm to solve the multicollinearity problem in the aeromagnetic compensation model (Leach, 1980). In recent years, the main methods to solve multicollinearity problems are the principal component analysis (Wu et al., 2018), the truncated singular value decomposition (TSVD) (Gu et al., 2013; Deng et al., 2013), the multi-model compensation method (Zhao et al., 2019), the wavelet analysis method (Deng et al., 2010; Dou et al., 2016), and the improved recursive least-squares (Zhao et al., 2017). The above methods are all based on linear models. In 1993, Williams proposed a neural network nonlinear model to solve aeromagnetic interference (Williams, 1993), but this neural network model has an overfitting problem; on



this basis, Ma Ming proposed the dual estimation compensation method of unscented Kalman filter, by introducing measurement noise, suppresses the problem of neural network overfitting (Ma et al., 2017). The measured value of the airborne magnetic sensor is the superposition of the geomagnetic field and interference field, which is usually separated by a high pass filter (Jia et al., 2004; Groom et al., 2004; Dou et al., 2016). However, due to the existence of a geomagnetic gradient, the filter can not completely separate the geomagnetic field. Besides, the induced magnetic field component and eddy current magnetic field component in the interference magnetic field is determined by the geomagnetic field. Therefore, there is a strong coupling relationship between the geomagnetic field and magnetic interference (Dou et al., 2016)).

The traditional fixed-wing platform produces interference from the helicopter platform, while the towed bird system is far away from the helicopter, so the interference generated by the helicopter in the towed bird is small and can be ignored. However, the measurement device and the electrified wire in the towed bird platform system will cause interference, so it is necessary to compensate for the interference of the towed bird platform. Because the towed bird system is affected by external factors, there are two modes of motion: large swing and small vibration. The geomagnetic gradient changes greatly in the large swing mode, so the interference of the geomagnetic gradient on the towed bird is an important factor. In this paper, based on the towed bird interference model, the geomagnetic gradient component is introduced and finally solved by the ridge estimation method. Through the actual flight data verification, this method can improve the data quality of pod interference.

## 2 Experiment and data introduction

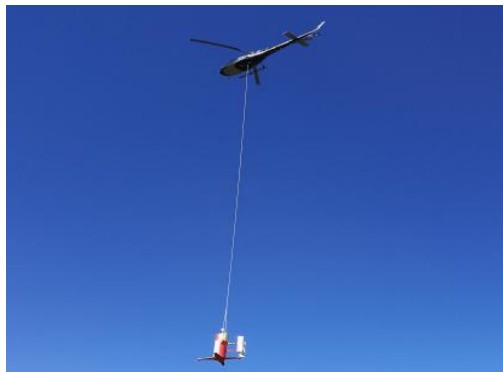

**Figure 1: Towed bird system**

In the process of aeromagnetic exploration, most methods use fixed-wing platforms to compensate. When the magnetic sensor is located near the fuselage or even inside the fuselage, the structure and changes of the interfering magnetic field generated by the aircraft in flight are quite complicated, which will also cause a large aeromagnetic interference (Xiu et al., 2018). In order to reduce the aeromagnetic interference, we used the helicopter towed bird method to conduct field test measurements in the Zhanhe area of Wudalianchi City, northern Heilongjiang Province. The towed bird is connected with the helicopter through a 30-meter long rope in Fig. 1. The interference generated by the helicopter in the towed bird is small,


and the influence on the measurement data of the optical pump magnetometer can be ignored. Because the ferromagnetic material in the helicopter towed bird system will affect the measurement data of the magnetic sensor, it is necessary to compensate for the magnetic interference generated by the towed bird system. There are two kinds of motion modes in the

motion process of the towed bird platform: one is the large amplitude swing mode influenced by the helicopter motion, the other is the small amplitude vibration mode influenced by the wind speed, and other factors in the swing process. Under the joint action of the two motion modes, the measured data have interfered. The experiment uses an optical pump sensor to measure the total magnetic field strength, and also uses a three-component fluxgate and an inertial navigation system to measure the attitude change of the pod. The sampling rate of the system is 10 Hz.

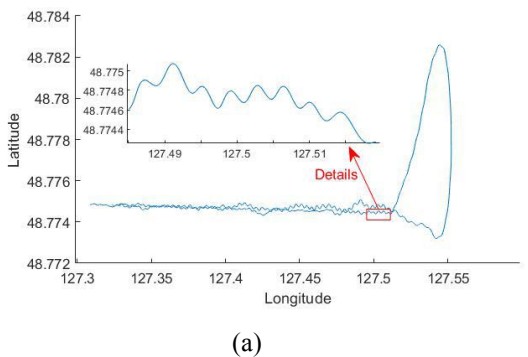

(a)

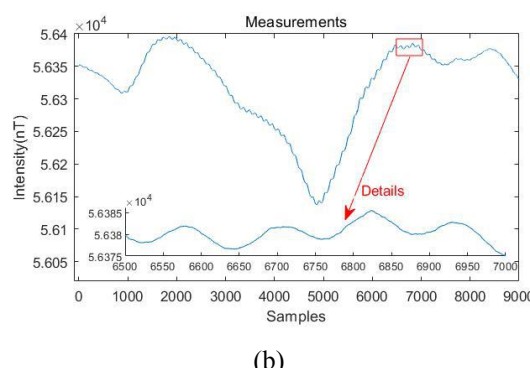

(b)

**Figure 2: Straight line and data**

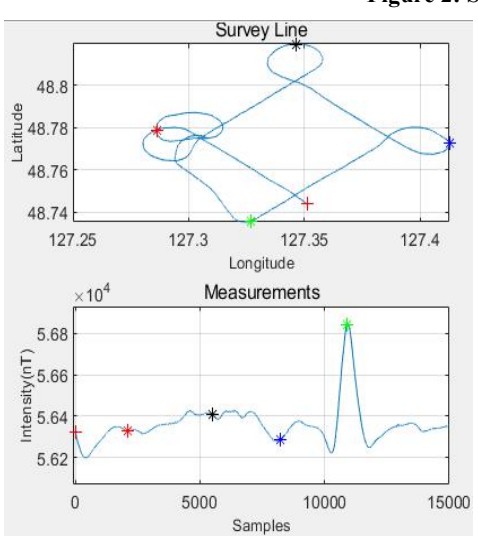

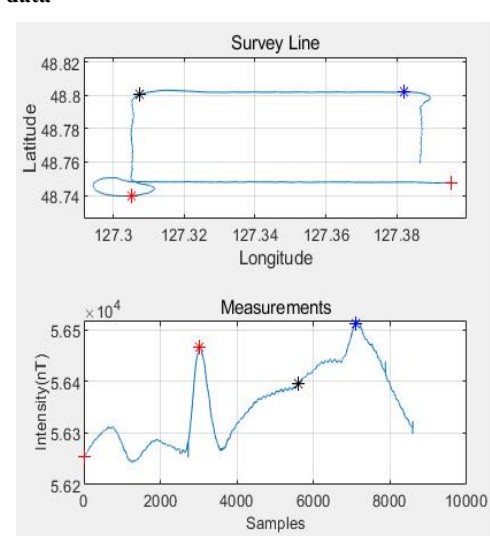

**Figure 3: Diamond line and data**          **Figure 4: Square line and data**

The experiment included three flights at the altitude of 1250m. The first time was a long straight flight. The purpose of the experiment was to observe the distribution of the geomagnetic field in the experimental area and the intensity of magnetic interference generated by the towed bird. Figure 2(a) shows the flight path, where the survey line direction corresponds to the measured value in Fig. 2(b). The range of latitude variation in Fig. 2(a) shows that the towed bird system platform has a



large swing range, and the measured value in Fig. 2(b) shows that the magnetic interference is about 5nT. The second flight
and the third flights are diamond and square flight routes respectively. In Fig. 3 the diamond data is used for the training of
the aeromagnetic compensation model, and in Fig. 4 the square data is used as the verification of the compensation effect of
the training model.

## 3 Compensation method

### 3.1 Towed bird model

The interference generated by the helicopter towed system and the fixed-wing helicopter platform is the interference
generated by the magnetic sensor strap-down system platform. The helicopter towed bird system interference is generated by
the towed bird platform, and the fixed-wing interference is generated by the helicopter platform.

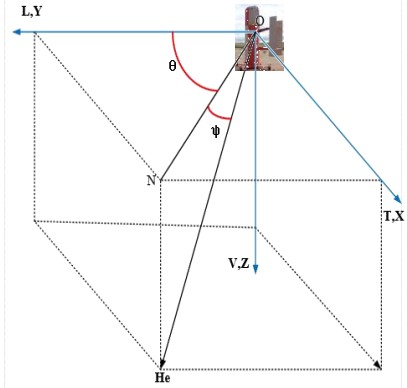

**Figure 5: The towed bird coordinate system**

The establishment of the towed bird coordinate system is shown in Fig. 5, where T, L, and V are the three coordinate axes
of the towed bird coordinate system. The side of the selected towed bird system is the left side parallel to the T axis, and as
the positive direction, The selected left side is the reference and rotated 90° clockwise and parallel to the L axis, and as the
positive direction, the helicopter of the vertical towed bird is the V axis, and downward is the positive direction. $X,Y$ and $Z$
are the angles between the three coordinate axes of the towed bird and the geomagnetic field, called Euler angles, where $\theta, \psi$
are the geomagnetic declination and the geomagnetic inclination, respectively.

Euler angles $X,Y,Z$ can be measured by a three-axis magnetometer, and are redefined as follows:

$$u_1 = \cos(X)$$
$$u_2 = \cos(Y) \tag{1}$$
$$u_3 = \cos(Z)$$

$u_1, u_2, u_3$ represents the direction cosine of the Euler angle.


The magnetic field generated by the towed bird system platform can be divided into three categories: permanent magnetic field, induced magnetic field, and eddy current magnetic field. Refer to the T-L model and establish the towed bird interference platform model as follows:

$$h_I = x_1 u_1 + x_2 u_2 + x_3 u_3 + x_4 u_1^2 + x_5 u_1 u_2 + x_6 u_1 u_3 + x_7 u_2 u_3 + x_8 u_3^2 + x_9 u_1 (u_1)' + x_{10} u_1 (u_2)' +$$
$$x_{11} u_1 (u_3)' + x_{12} u_2 (u_1)' + x_{13} u_2 (u_3)' + x_{14} u_3 (u_1)' + x_{15} u_3 (u_2)' + x_{16} u_3 (u_3)' \tag{2}$$

$h_I$ is aeromagnetic interference, $x_i, i = 1,2,3...16$ is the aeromagnetic interference parameter, $u_1', u_2', u_3'$ is the derivative of Euler angle cosine to time.

Further expressed as:

$$h_I = u \cdot x \tag{3}$$

Where $x = [x_1 \, x_2 \, x_3 ......x_{16}]^T$ is the aeromagnetic interference parameter, and $[u = u_1 \, u_2 \, u_3 ......u_3(u_3)']^T$ is the aeromagnetic interference feature.

The measured value $h$ can be expressed by the linear superposition of the geomagnetic field and aeromagnetic interference as:

$$h = H_e + h_I = H_e + u \cdot x \tag{4}$$

**3.2 Error analysis and improvement**

The above analysis of the towed bird model shows that the aeromagnetic measurement value is a linear superposition of the geomagnetic field and the aeromagnetic interference. Traditional aeromagnetic compensation usually uses a 0.1-0.6Hz bandpass filter to obtain the aeromagnetic interference value [5]. $bpf(R)$ is expressed as linear bandpass filtering for each column of matrix R. applying $bpf$ to both ends of formula (4), we can get the following results:

$$bpf(h) = bpf(H_e + bpf(u \cdot x)) = bpf(H_e) + bpf(u) \cdot x \tag{5}$$

When $bpf(H_e) = 0$, then:

$$bpf(h) = bpf(u) \cdot x \tag{6}$$

The data processed by the bandpass filter are recorded as $h_f, u_f$. Let $bpf(h) = h_f, bpf(u) = u_f$. Then (6) can be expressed as:

$$h_f = u_f \cdot x \tag{7}$$

Applying the least-squares to the previous test to solve:

$$x = (u_f^T u_f)^+ u_f^T h_f = u_f^+ h_f \tag{8}$$

Where $(u_f^T u_f)^{-1} u_f^T$ is the generalized inverse of the matrix $u_f$, denoted as $u_f^+$.

By analyzing the influence of the singular value of the system matrix $u_f$ on the least-squares solution, the singular value decomposition (SVD) of the filtered feature matrix $u_f$ is obtained:

$$u_f = USV^T = \sum_{i=1}^{n} u_{Ai} \sigma_{Ai} v_{Ai} \tag{9}$$


Where, the matrix $u_f$ is $N \times 16$ matrix, $U$ and $V$ are $N \times N, 16 \times 16$ orthogonal matrix, $S$ is $n \times 16$ diagonal matrix, $\sigma_{Ai}$ is

the ith singular value and has $\sigma_{A1} \geq \sigma_{A2} \ldots \geq \sigma_{An}$; $u_{Ai}$ is the ith column vector of matrix $U$; $v_{Ai}$ is the ith column vector of matrix $V$.

Then, the least-squares solution (8) can be expressed as follows:

$$x = \sum_{i=1}^{n} \frac{u_{Ai}^T h_f}{\sigma_{Ai}} v_{Ai} \tag{10}$$

It can be obtained from the above formula that the system matrix $u_f$ contains very small singular values, the slight error in

the matrix $h_f$ will be magnified. Because the helicopter towed bird system has a large change range, the geomagnetic gradient component will be introduced, which will cause the geomagnetic field frequency and the aeromagnetic interference frequency to be mixed and superimposed, resulting in the presence of a geomagnetic field component in the filtered aeromagnetic interference $h_f$, which leads to a large error in the solution.

From the above analysis, it can be known that the existence of the geomagnetic gradient will affect the compensation

result of the towed bird interference. According to the two-dimensional Taylor model of the geomagnetic field, the horizontal geomagnetic field is expressed as a function of latitude and longitude $y$(Dawson and Newitt, 1977).

$$(H_e)_{hor} = \sum_{n=0}^{N} \sum_{k=0}^{n} c_{nk} (a - a_0)^{n-k} (y - y_0)^k \tag{11}$$

Where $(H_e)_{hor}$ is the horizontal geomagnetic field. $a_0$ and $y_0$ are the latitude and longitude of the initial position, $c_{nk}$ is a constant, and N is the truncation order.

The filtered horizontal geomagnetic field obtained through band-pass filter processing is as follows:

$$bpf((H_e)_{hor}) = bpf(\sum_{n=0}^{N} \sum_{k=0}^{n} c_{nk} (a - a_0)^{n-k} (y - y_0)^k) \tag{12}$$

The Taylor model only considers the relationship between latitude and longitude and the geomagnetic field and does not consider the impact of altitude changes on the geomagnetic field. Since the helicopter's flying height is 1500 meters, it is considered that the vertical geomagnetic field gradient is proportional to the helicopter's flying height. Assuming that the

scale factor of the vertical gradient component of the geomagnetic field is c, the filtered vertical gradient component can be expressed as follows:

$$bpf((H_e)_{ver}) = bpf(c(z - z_0)) = cbpf(z) \tag{13}$$

Where $bpf((H_e)_{ver})$ is the filtered vertical geomagnetic field gradient value, $z_0$ is the height of the starting position of the towed bird, and $z$ is the height of the towed bird during flight. Then the geomagnetic field passing through the band-pass

filter can be further expressed as follows:

$$bpf(H_e) = bpf(\sum_{n=0}^{N} \sum_{k=0}^{n} c_{nk} (a - a_0)^{n-k} (y - y_0)^k) + cbpf(z) \tag{14}$$

When the truncation order N of the local magnetic field is different, the expression of the two-dimensional Taylor model of the geomagnetic field is different, which will affect the final compensation result.

Next, the truncation order N=1,2,3,4 is sorted as follows through the bandpass filter:

$N = 1$ $bpf((H_e)_{hor1}) = c_{10}bpf(a) + c_{11}bpf(y)$


$$N = 2 \quad bpf((H_e)_{hor2}) = c_{10}bpf(a) + c_{11}bpf(y) + c_{20}bpf(a^2) + c_{22}bpf(y^2) + c_{21}bpf(ay)$$

$$N = 3 \quad bpf((H_e)_{hor3}) = c_{10}bpf(a) + c_{11}bpf(y) + c_{20}bpf(a^2) + c_{21}bpf(ay) + c_{22}bpf(y^2) + c_{30}bpf(a^3)+$$
$$c_{31}bpf(a^2y) + c_{32}bpf(ay^2) + c_{33}bpf(y^3)$$

$$N = 4 \quad bpf((H_e)_{hor4}) = c_{10}bpf(a) + c_{11}bpf(y) + c_{20}bpf(a^2) + c_{21}bpf(ay) + c_{22}bpf(y^2) + c_{30}bpf(a^3)+$$
$$c_{31}bpf(a^2y) + c_{32}bpf(ay^2) + c_{33}bpf(y^3) + c_{40}bpf(a^4) + c_{41}bpf(a^3y) + c_{42}bpf(a^2y^2) + c_{43}bpf(ay^3)+c_{44}bpf(y^4)$$

By introducing formula (11) into formula (5), we can get the following results:

$$bpf(h) = bpf(H_e + bpf(u \cdot x)) = bpf(H_e) + bpf(u) \cdot x = bpf((H_e)_{hor}) + cbpf(z) + bpf(u) \cdot x \quad (15)$$

$bpf((H_e)_{hor})$ can bring in different results according to different values of N in equation (14), and finally combine the towed bird model with the geomagnetic field model, and the final expression is as follows:

$$bpf(h) = bpf(u_\theta)x_\theta \quad (16)$$

Where $u_\theta = [u,z,a,y,a^2 ...], x_\theta = [x,c,c_{10},c_{11},c_{20}...]$. $u_\theta$ and $x_\theta$ have different expressions according to the value of N. For example, when N=1, $u_\theta = [u,z,a,y]$, $x_\theta = [x,c,c_{10},c_{11}]$。

Because the geomagnetic field component is introduced into the T-L model, the model will further have the problem of complex collinearity. Therefore, the ridge estimation method is introduced to solve the problem. The ridge estimation solution formula is as follows:

$$\widehat{x_\theta} = \underset{x_\theta}{\mathrm{argmin}} \left( ||bpf(h) - bpf(u_\theta)x_\theta||^2 + \lambda||x_\theta||^2 \right) \quad (17)$$

Among them, $\widehat{x_\theta}$ is the parameter estimation value under the ridge estimation, and $\lambda$ is the regularization factor.

**3.3 Evaluation standard of compensation quality**

The traditional aeromagnetic compensation quality evaluation standard uses a standard deviation improvement ratio to evaluate:

$$IR = \frac{\sigma_{before}}{\sigma_{after}} \quad (18)$$

$\sigma_{before}$ and $\sigma_{after}$ respectively are the standard deviation of the data before and after compensation. Because the standard deviation data includes not only aeromagnetic interference data but also geomagnetic gradient data. Assuming that the aeromagnetic interference and the geomagnetic field can be linearly superimposed, then:

$$IR = \frac{\sigma_{before}^I+\sigma_{before}^E}{\sigma_{after}^I+\sigma_{after}^E} \quad (19)$$

$\sigma_{before}^I$ and $\sigma_{after}^I$ are the standard deviations of the aeromagnetic interference before and after compensation, $\sigma_{before}^E$ and $\sigma_{after}^E$ are the standard deviations of the geomagnetic field before and after compensation. However, the data used in this paper is towed bird data. Because of the small aeromagnetic interference generated by the bird, the geomagnetic gradient is larger and the geomagnetic gradient changes little before and after compensation.

$$\sigma_{before}^I \ll \sigma_{before}^E, \quad \sigma_{after}^I \ll \sigma_{after}^E, \quad \sigma_{before}^E \approx \sigma_{after}^E \quad (20)$$


Therefore:

$$IR = \frac{\sigma^I_{before} + \sigma^E_{before}}{\sigma^I_{after} + \sigma^E_{after}} \approx \frac{\sigma^E_{before}}{\sigma^E_{after}} = 1 \tag{21}$$

Therefore, the data before and after compensation is filtered by a high filter, and the cut-off frequency is 0.03 Hz. Then there are:

$$IR_{0.03} \approx \frac{\sigma^I_{before}}{\sigma^I_{after}} \tag{22}$$

The paper takes $IR_{0.03}$ as the evaluation index of the compensation result.

## 4 Results and analysis

According to the above analysis, when the truncation order of the two-dimensional Taylor model of the local magnetic field is $N = 0,1,2,3,4$, the ridge estimation method are used to solve the formula (17), and the standard deviation improvement ratio (IR) of formula (22) is used to evaluate the results of aeromagnetic compensation.

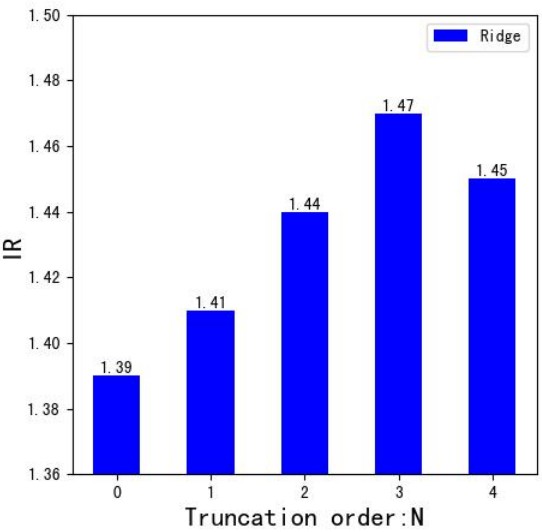

**Figure 6: Improved ratio (IR)**

Figure 6 shows the standard deviation improvement ratio of applying the ridge estimation method when the truncation order N is 0,1,2,3,4 respectively. The paper selects the truncation order of the two-dimensional Taylor of the geomagnetic field. When $N = 0$, the compensation result is less than the standard deviation improvement ratio when N is 1, but when N is 1, it is less than the standard deviation when N is 2 and 3. The improvement ratio shows that the linear function of the



geomagnetic gradient cannot truly reflect the geomagnetic field component. When N is 4, it will be slightly lower than the standard deviation improvement ratio when N is 3. When the truncation order is greater than 3, the multicollinearity of the

model will be increased, leading to the introduction of errors in the solution process, so choosing a suitable truncation order is very important for model solving. When N is 3, the ridge estimation method is used to solve the problem, and the final compensation result is the best, and the standard deviation improvement is 6% higher than that of the compensation effect without the introduction of the geomagnetic gradient.

Figure 7 shows the comparison of the standard deviation and improvement ratio of the towed bird compensation in

different directions. Figure 8 shows the comparison of the original measurement value and the compensation result when N=1,3. It can be seen from Fig. 7 and Fig. 8 that when the helicopter flies in the south and west directions, the standard deviation is large, the swing of the towed bird is small, and the interference is mainly caused by the vibration mode of the towed bird. Therefore, when the geomagnetic gradient is introduced into the compensation, the result is only slightly better than the model when $N = 1$. When the helicopter is heading north, the towed bird platform is affected by large swings and

vibration effects, resulting in greater aeromagnetic interference, and is greatly affected by the geomagnetic gradient. Introducing the geomagnetic gradient into the towed bird interference model will be greatly improved, and IR will be

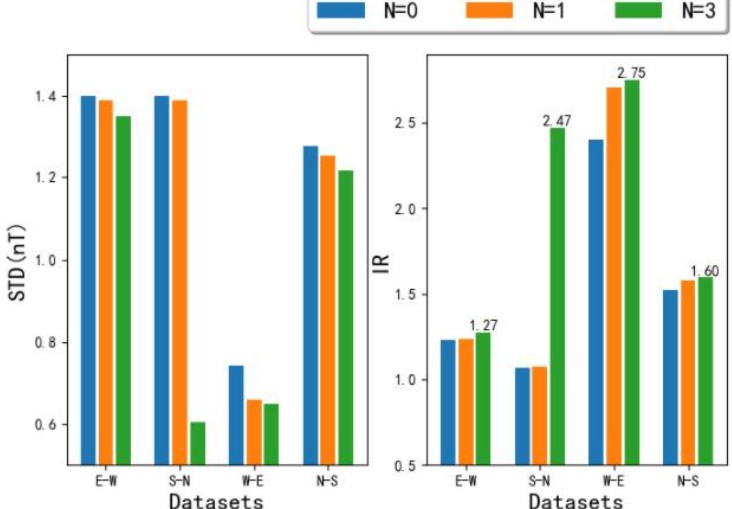

**Figure 7: Comparison of compensation results of N = 0,1,3 in different directions:**
**(a)standard deviation (STD),and(b)improvement ratio (IR).**





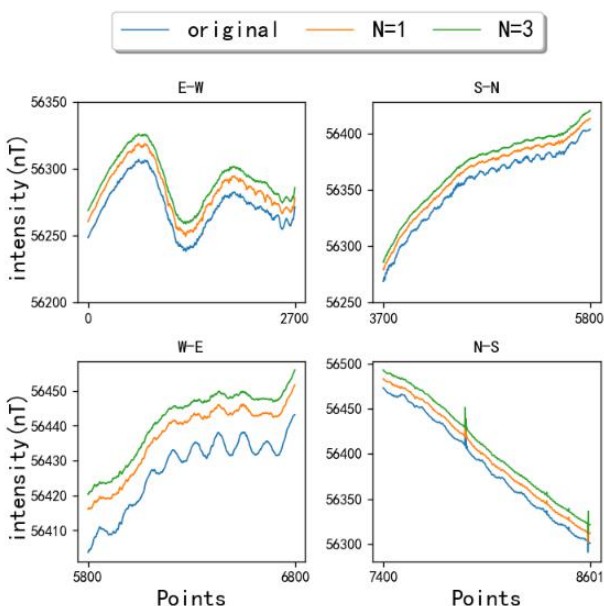


**Figure 8 The original value is compared with the result after $N = 1,3$ compensation**

improved to 2.47. When the helicopter is heading east, the interference is mainly caused by the swing mode of the towed bird, the standard deviation interference is small, and it is greatly affected by the geomagnetic gradient, and the IR is increased to 2.75.

**5 Conclusion**

The paper analyzes the two movement modes of the towed bird system during the movement process, considers the influence of the changing geomagnetic gradient on the results of magnetic interference compensation, introduces the changed geomagnetic gradient into the interference model, and deduce the parameter estimation and correction of geomagnetic gradient on the model. The paper solves the influence of changing geomagnetic gradient on the compensation

results under the towed bird system, and further expands the towed bird interference model. When the towed bird system is subject to large swings and vibrations in the heading, this method can improve the data quality of aeromagnetic interference,the experimental results show that the improvement ratio has increased by 6%. Next, we will use this compensation method to improve the data quality of aeromagnetic surveys and use the helicopter towed bird system to detect underground magnetic targets.


**Author Contributions:** Zhijian Zhou and Zhilong Liu. designed the experiments and wrote the paper; Wenduo Li helped to discuss the research and modified the document; Yihang Wang performed the experiments and analyzed the data; Chao Wang led the project and provided support. All authors have read and agreed to the published version of the manuscript.



**Data Availability Statement:** The data presented in this study are available on request from the corresponding author. The
data are not publicly available due to permissions.

**Conflicts of Interest:** The authors declare no conflict of interest.

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
