# Peer review of "Analysis and reduction for the influence of geomagnetic gradient on aeromagnetic compensation in a towed bird"

_Geoscientific Instrumentation, Methods and Data Systems, 2021_

## Referee Comment (RC1)

[revised manuscript text omitted]

Where $u_0 = [u,z,a,y,a^2...], x_0 = [x,c,c_{10},c_{11},c_{20}...].$ $u_0$ and $x_0$ have different expressions according to the value of N. For

example, when N=1, $u_0 = [u,z,a,y],$ $x_0 = [x,c,c_{10},c_{11}].$

Because the geomagnetic field component is introduced into the T-L model, the model will further have the problem of

complex collinearity. Therefore, the ridge estimation method is introduced to solve the problem. The ridge estimation

170 solution formula is as follows:

[revised manuscript text omitted]

---

## Author Response (AR1)

Dear RC1:

We are very grateful to this referee comments, and we have carefully read and considered the referee's comments, and these comments are important for improving the quality of this manuscript. Based on these comments, we have made carefully modification and proofreading on the original manuscript.

Best regards,

Zhijian Zhou.

This paper is about aeromagnetic noise compensation of a total-field magnetometer mounted in a bird towed by a helicopter. The paper suggests that the Tolles-Lawson noise model can be improved upon by introducing x,y,z position terms to the ridge regression solution. Overall the paper is interesting but I am uncertain if it is novel as the survey method has been around since the mid 20[th] century and introducing position terms has been around for just as long. I would ask that they highlight what is novel in this paper.

The paper first analyzes the influence of geomagnetic gradient on the results of aeromagnetic compensation. Secondly, we model the geomagnetic field according to the position information. According to the Taylor model, the different truncation orders have different effects on the results. When the order is 1,2, the real geomagnetic field cannot be simulated. When the order is greater than 3, the multicollinearity of the model will be increased and the solution result will be affected. Therefore, the paper finally determines that when the truncation order is 3, there is the best compensation effect.

Their reasoning using analytical expressions makes sense but I believe could be made more concise. The use and reuse of certain variables is confusing. Their results presented do suggest reduced noise, although I think they could be a little more thorough. See comments below.

The paper needs a good rewrite. I find many sentences confusing and have to guess their meaning. I would recommend a major revisions.

Thanks again for your comments. I will re-write the paper.

Specific comments:

-some paragraphs are indented, other are not

Thanks, I made the change

-(line 23), its "Bickel"

Changed

-(line 39, 62, 73, 183, 203) contains a sentence that does not make any sense. These are just examples of many.

I changed and deleted (line 39, 62, 73, 183, 203)

-(line 46 and throughout), I believe the better term is "ridge regression" not "ridge estimation"

Thanks, I made the change

-(line 56) can you support that the interference generated by the helicopter is small? Using the TL coefficients, can you estimate where the bulk of the noise of the bird is coming from? Ie permanent, induced magnetization or eddy current。

We have conducted a magnetic field measurement. When the towed bird system is 30 meters away from the helicopter, the interference is less than 0.1nT, so it can be ignored. The bulk of the noise of the bird is coming from the ferromagnetic substance in the towed bird. The interference include constant magnetic fields, induced magnetic fields and eddy current magnetic fields.

-(Figure 3), was the diamond line data used in the results? If not, it should be removed. Or results should be added. I would prefer the latter as the results are sparse and barely convincing.

We use diamond data for model training data, and orientation data as a test of training results. Only in this way can the results of the model be tested.

-(line 70) the altitude of the helicopter or bird was 1250m? Above ground or GPS height? (Above ground is really what matters). Why does line 143 say 1500m? How far below the helicopter was the bird? What is the std of the earth's field at this altitude? Or if the earth's field is the concern why wasn't the results collected at lower altitude close the ground where variations are greater?

The height of the bird is 1250 meters, measured by GPS. Due to negligence, the height of line 143 was modified. The bird is 30 meters below the helicopter. The standard of the earth's magnetic field is 56300nT. Due to the use of helicopters for flight, for flight safety considerations, the flight altitude is 1250 meters.

-(lines 85-90) this can be better explained with a diagram, or amend figure 5, or just reference another paper that explains it. I believe leliak covers this well along with much of this section.

Since this paper uses the towed bird as the carrier, I will re-explain the establishment of this part of the carrier coordinate system. I rewritten this part of the content and introduced references.

-(equation 2) remove the brackets

The brackets removed

-(line 102) both matrices cannot be transposed for proper multiplication

I removed the transposition of the formula u,$u = [u_1 \ u_2 \ u_3 ......u_3 (u_3)^{'}]$

-(line 100) reference 5 does not exist, references are not numbered. Also I do not agree that the 0.1-0.6Hz band is the typical interference band. Also sentence needs to be worded better.

Corrected the errors in the paper.

-(line 121) explain why SVD is needed.

SVD can be used to obtain the influence of the eigenvalue on the result:For example, equation (10) $x = \sum_{i=1}^{n} \frac{u_{Ai}^T h_f}{\sigma_{Ai}} v_{Ai}$.When the singular value $\sigma_{Ai}$ has a small value, the error in $h_f$ will have a greater impact. Therefore, we use SVD for error analysis.

-(line 124)  what is N? what is n?

Since N is used below in the paper, and N and n have been corrected, R is used instead to indicate the number of sampling points.

-(equation 10) I believe there is an error in this equation when you substitute equation 9 into it. Sigma should be in the numerator

Equation (10) is not only substituting equation (9), but also the pseudo-inverse of equation (9). So I think it is correct, and thank you for your comments.

-(line 130) "large change range" requires more explanation

The towed bird system cannot fly as standard. Because the bird system is affected by wind during the flight, the maximum angle of the bird reaches 25 degrees, so it is called the large range of change.

 -(equation 11 and variables throughout) I would suggest changing the latitude variable to x. change all x variables to c or c_something because it is a coefficient. This will clean up line 167 where you have a mix-mash of variables.

I changed the latitude variable to x and all the x variables to c.

-(equation  12) you reuse N and n, I would avoid this and just use different letters

I replaced n with m in the paper.

-(line 136) Dawson and Newitt fit the components, not the horizontal gradient. They also used 6th order polynomial. Further explanation is needed as to why equation 11 is valid.

Dawson and Newitt fit the components are only related to the latitude and longitude of the geomagnetic field, not the height; therefore, we think that the horizontal geomagnetic field is Dawson and Newitt fit the components. Because the order is different, it will cause the difference in the results, so we discuss N as a different result.

-(line 152) equation 14 has 3 dimensions, lat, long and z

equation 14 has 3 dimensions, latitude, longitude and z. The corresponding text has been changed.

-(line 161) equation reference 11 should be 14 I believe

The equation reference is 14, which has been corrected

 -(line 198) why do you switch the filter bank to 0.03Hz specifically? After say the band should be 0.1-0.6Hz? Is the upper limit still 0.6Hz?

Due to my negligence, I have changed the sentence that the band pass filter is 0.1-0.6 Hz. We use a bandpass filter of 0.03-0.2Hz.

-(line 195) font size change

The change is completed.

-what ridge regression coefficient do you use?

When N takes different values, the values of the ridge    regression parameters are different as shown in Table 1.

Table 1. The ridge regression parameters

| N | 0 | 1 | 2 | 3 | 4 |
|---|---|---|---|---|---|
| coefficient | 0.003 | 0.0012 | 0.00072 | 0.000052 | 0.00000022 |

-(Figure 6) can you calculate the error on your IR calculations. Are you sure the differences are significant? Can you should the trend between N:1-6? Figure 6 should look like Figure 7. The caption should state the difference in results between Figure 6 and 7 because I am unclear why Figure 6 results are not included in Figure 7. Perhaps they should be combined? Regardless, further clarity is needed here.

Since IR is the improvement ratio of the standard deviation of the data before and after compensation, the larger the IR, the better the compensation result. When N:1-3 IR increases, when N:4-6, IR decreases. When N=3 relative to N=0, IR is increased by 6%. It shows that this method can further compensate for aeromagnetic interference. Figure 6 is the entire direction route compensation. In order to analyze the influence of this method on the compensation results in each direction, we have drawn Figure 7.

-(line 213) what evidence do you have that the noise is a vibration mode of the towed bird? Or do you mean swing?

Through the analysis of actual measurement data, the interference comes from the two modes of vibration and swing of the towed bird. The swing will introduce large-angle change interference, and the vibration is a small-angle interference. Two interference modes constitute the interference of the towed bird.

-(line 216) "greatly improved" may be an overstatement when looking at Figure 6. Or explain why.

A slight improvement, the improvement ratio is 6%.

-(Figure 7) Why remove N=2? Is the STD the compensated or uncompensated signal?

Added the result of N=2, STD is the result of the compensation signal, which has been noted in the annotation in the picture.

-(Figure 7) VERY IMPORTANT. Why is the S-N noise reduced so significantly for N=3 compared to N=1? Could this point towards a much better noise model? Or is something else going on here?

Because when the aircraft is flying in the S-N direction, the angle of the towed bird changes is greater than in other directions, leading to a large error in the introduction of the geomagnetic field. When N=3, the geomagnetic field can be simulated better, so choosing N=3 can suppress the influence error of the geomagnetic field change, and there is a better improvement result in S-N.

-(line 250/254, and other instances in the references) inconsistent naming of B.W. Leach. See other errors in the scanned copy. More thorough review is needed.

Make consistent changes to the naming of B.W. Leach. in the references.

Thanks for your comment.
The title should be reformulated –(analysis and reduction?)
Title changed: Analyze and reduction the influence of geomagnetic gradient on aeromagnetic compensation in a towed bird

L. 39 "The traditional fixed-wing platform produces interference from the helicopter platform" – unclear statement.
This sentence was deleted.

L.43-44 "The geomagnetic gradient changes greatly in the large swing mode"- How large is a swing? A few meters? Does geomagnetic gradient change noticeably along this distance? Explanations are needed.
The swing amplitude is 10 meters. The geomagnetic gradient is 0.5nT/m, I will re-explain it in the data introduction below.

L.61 "other factors in the swing process." What do you mean?

The imperfections of this sentence have been changed.

L. 83 "fixed-wing interference is generated by the helicopter platform." –This is in contradiction with your statement in L. 56 "The interference generated by the helicopter in the towed bird is small and .. can be ignored".

This paper uses the towed bird system. The sensor is 30 meters away from the helicopter, so the interference from the helicopter can be ignored. For fixed-wing aircraft platforms, the sensors are fixed on the aircraft, so fixed-wing interference is generated by the helicopter platform

L. 103-104 "is the aeromagnetic interference feature." – u is the combination of direction cosines and their derivatives. – Should be reformulated.

Changed this sentence:$[u = u_1\, u_2\, u_3 ......u_3 (u_3)']^T$ is the the aeromagnetic interference feature, which is the combination of direction cosines and their derivatives.

L.130 "helicopter towed bird system has a large change range" – What range? How large? Unclear!

helicopter towed bird system

During the movement, the swing distance is 10 meters. Because the geomagnetic gradient is 0.5nT/m, the geomagnetic gradient cannot be ignored.

L.140 "The filtered horizontal geomagnetic field obtained through band-pass filter processing is as follows" – Optical pumping magnetometer is a total field instrument. It does NOT measure horizontal component of geomagnetic field.

The horizontal geomagnetic field is obtained through latitude and longitude modeling. It is not measured by an optical pump magnetometer.

L. 179 –" aeromagnetic interference and the geomagnetic field can be linearly superimposed" -However in L.38 you mentioned "there is a strong coupling relationship between the geomagnetic field and magnetic interference"

The T-L model considers the geomagnetic field to be constant, which is obtained by linear superposition. In reality, the geomagnetic field changes, so there is a coupling relationship between the geomagnetic field and magnetic interference. This paper addresses the influence of changing geomagnetic field on aeromagnetic interference.

L.183-184 "Because of the small aeromagnetic interference generated by the bird, the geomagnetic gradient is larger and the geomagnetic gradient changes little before and after compensation"- Unclear! Larger than what is geomagnetic gradient?

The maximum difference of the geomagnetic field gradient is 250nT in figure 1, and the geomagnetic field gradient changes very little before and after compensation, while the aeromagnetic interference value is only 5nT. Therefore, this paper uses a 0.03Hz high-pass filter to obtain the aeromagnetic interference value, and evaluates the improvement ratio of the standard deviation before and after compensation.

[Figure]

**Figure1 Data**

L.210-215 Why there is a difference in south and west flight directions relative to north and east?

I have not found any comparison of Tolles Lawson coefficients obtained by means of suggested method relative to standard procedure. Does this new technique make sense?

Since this paper uses a helicopter pod system, it is affected differently by wind in different directions. When flying northward, the swing is less affected by the wind, so the compensation effect is better. When N=0, it is the traditional T-L model. This paper compares the solution with the traditional T-L model in Figure 2. Although the improvement ratio has only increased by 6%, I believe it is an improvement in aeromagnetic compensation.

[Figure]

Figure 2: Improved ratio (IR)

Quality of English language is poor.

Thank you for your comments, I re-written the paper.

---

## Author Response (AR2)

The authors revised the MS - it looks better.

However, Figures are of low quality yet.

Thanks for your comment. I completed the changes to the pictures.

List of References is too short. Some references are too old.

I replaced and added related references.

What is "towed bird" in the title - understand only narrow specialists.

The towed bird is a professional term. It connects the magnetic sensor to the helicopter through a rope.

The towed pod is used in the following references. I explained the towed bird system in the experiment.

Kaneko, T., et al. "Low-Altitude Remote Sensing of Volcanoes using an Unmanned Autonomous Helicopter: An Example of Aeromagnetic Observation at Izu-Oshima Volcano, Japan." International Journal of Remote Sensing, vol. 32, no. 5, 2011, pp. 1491-1504.

Kratzer, Terence, and Julian Vrbancich. "Real-Time Kinematic Tracking of Towed Aem Birds." Exploration Geophysics (Melbourne), vol. 38, no. 2, 2007, pp. 132-143.

Xiu, Chunxiao, et al. "Compensation for Aircraft Ef Davis, Aaron C., James Macnae, and Terry Robb." Pendulum Motion in Airborne HEM Systems." Exploration Geophysics (Melbourne), vol. 37, no. 4, 2006, pp. 355-362.

Xiu, Chunxiao, et al. "Compensation for Aircraft Effects of Magnetic Gradient Tensor Measurements in a Towed Bird." Exploration Geophysics (Melbourne), vol. 49, no. 5, 2018;2017, pp. 713-725.

So, intermediate revision is necessary.

---

## Author Response (AR3)

Due to my negligence, I changed the title of the paper again.

**Analysis and reduction for the influence of geomagnetic gradient on aeromagnetic compensation in a towed bird**